

# Transcriptome analysis of *Chelidonium majus* elaiosomes and seeds provide insights into fatty acid biosynthesis

Jiayue Wu[1], Linlin Peng[1], Shubin Dong[2], Xiaofei Xia[3] and Liangcheng Zhao[1,4]

[1] College of Nature Conservation, Beijing Forestry University, Beijing, China
[2] College of Biological Sciences and Technology, Beijing Forestry University, Beijing, China
[3] Beijing Museum of Natural History, Beijing, China
[4] Museum of Beijing Forestry University, Beijing Forestry University, Beijing, China

## ABSTRACT

**Background:** Elaiosomes are specialized fleshy and edible seed appendages dispersed by ants. Lipids are the primary components of elaiosomes. *Chelidonium majus* is a well-known plant, the seeds of which are dispersed by ants. Previous studies have identified the presence of primary fatty acids in its elaiosomes and seeds. However, the molecular mechanisms underlying fatty acid biosynthesis in elaiosomes remain unknown.

**Methods:** In order to gain a comprehensive transcriptional profile of the elaiosomes and seeds of *C. majus*, and understand the expression patterns of genes associated with fatty acid biosynthesis, four different developmental stages, including the flower-bud (Ch01), flowering (Ch02), young seed (Ch03), and mature seed (Ch04) stages, were chosen to perform whole-transcriptome profiling through the RNA-seq technology (Illumina NGS sequencing).

**Results:** A total of 63,064 unigenes were generated from 12 libraries. Of these, 7,323, 258, and 11,540 unigenes were annotated with 25 Cluster of Orthologous Groups, 43 Gene Ontology terms, and 373 Kyoto Encyclopedia of Genes and Genomes pathways, respectively. In addition, 322 genes were involved in lipid transport and metabolism, and 508 genes were involved in the lipid metabolism pathways. A total of 41 significantly differentially expressed genes (DEGs) involved in the lipid metabolism pathways were identified, most of which were upregulated in Ch03 compared to Ch02, indicating that fatty acid biosynthesis primarily occurs during the flowering to the young seed stages. Of the DEGs, acyl-ACP thioesterases, acyl carrier protein desaturase (*DESA1*), and malonyl CoA-ACP transacylase were involved in palmitic acid synthesis; stearoyl-CoA desaturase and *DESA1* were involved in oleic acid synthesis, and acyl-lipid omega-6 desaturase was involved in linoleic acid synthesis.

# INTRODUCTION

Elaiosomes are highly specialized nutrient-rich structures that are attached to the seeds of many plant species. The term elaiosome was introduced for the first time by

Corresponding author
Liangcheng Zhao,
lczhao@bjfu.edu.cn

*Sernander (1906)* to refer to all fleshy and edible appendages of the diaspore dispersed by ants. Seed dispersal by ants is known as myrmecochory, a worldwide phenomenon found in a diverse range of habitats, including arid, tropical, and temperate regions (*Van Der Pijl, 1982*; *Werker, 1997*; *Giladi, 2006*). A recent study estimated that more than 11,000 species in 334 genera (about 2.5% of all genera) and 77 families (around 17% of all families) of angiosperm plants could be dispersed via myrmecochory (*Lengyel et al., 2010*).

Myrmecochory is regarded as a mutualistic relationship that is beneficial for both plants and ants. Ants typically collect and transfer whole diaspores (seed + elaiosome) to their nests, wherein the elaiosomes are consumed and the seeds are discarded unharmed within or outside the nests, which are in turn generally regarded to provide optimal germinating conditions (*Beattie, 1985*; *Hughes & Westoby, 1990*; *Giladi, 2006*). The elaiosomes can be fed to the larvae, which in turn promotes ant colony reproduction, fitness, and gyne production (*Morales & Heithaus, 1998*; *Bono & Heithaus, 2002*; *Fischer et al., 2005*; *Gammans, Bullock & Schönrogge, 2005*). Therefore, elaiosomes are key structures that increase the attractiveness of diaspores to ants (*Reifenrath, Becker & Poethke, 2012*). Elaiosomes develop from seed tissues (chalaza, funiculus, hilum, and raphe-antiraphe) through various mechanisms, but appear to have the same main function of attracting ants (*Lisci, Bianchini & Pacini, 1996*; *Morrone, Vega & Maier, 2000*; *Ciccarelli et al., 2005*; *Yang et al., 2015*). Hence, elaiosomes are considered suitable examples for studying convergent evolution in flowering plants (*Lengyel et al., 2010*).

Many studies have described several factors that likely determine the efficiency of myrmecochory. The sizes of seeds and elaiosomes, the elaiosome size/seed size ratio, and their physical surfaces influence the ability of ants to disperse the diaspores (*Oostermeijer, 1989*; *Hughes & Westoby, 1992*; *Mark & Olesen, 1996*). In addition to the morphological and structural characteristics of diaspores, chemical compounds present in the elaiosomes are additionally regarded as crucial factors that influence the removal of diaspores, considering that elaiosomes are beneficial for diaspore transport and food acquisition in ants (*Hughes, Westoby & Jurado, 1994*; *Fischer et al., 2008*; *Reifenrath, Becker & Poethke, 2012*; *Chen et al., 2016*). Although protein and starch are also present in elaiosomes, lipids generally constitute the major class of compounds in elaiosomes. Based on the prominent hypothesis that certain fatty acids induce the collection of diaspores by ants, most studies have focused on the fatty acids of the elaisomes and compared their compositions with those of seeds or even insects (*Marshall, Beattie & Bollenbacher, 1979*; *Soukup & Holman, 1987*; *Kusmenoglu, Rockwood & Gretz, 1989*; *Lanza, Schmitt & Awad, 1992*; *Hughes, Westoby & Jurado, 1994*; *Morrone, Vega & Maier, 2000*; *Boulay, Coll-Toledano & Cerdá, 2006*). A more recent chemical analysis of the diaspores of 13 central European ant-dispersed plant species from seven families revealed that elaiosomes have a more homogeneous fatty acid composition, with oleic acid as the most abundant fatty acid component, followed by palmitic acid and linoleic acid (*Fischer et al., 2008*).

*Chelidonium majus* L., commonly known as greater celandine, is the only species of the genus *Chelidonium* in the Papaveraceae family. *C. majus* is an herbaceous perennial

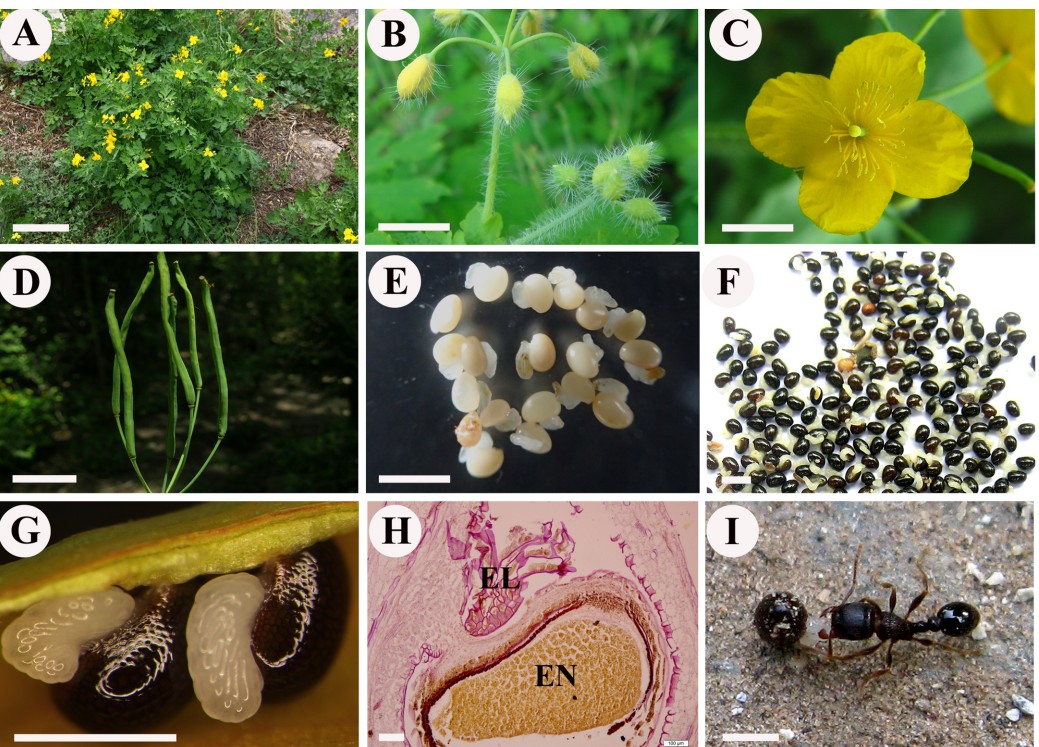

**Figure 1 Chelidonium majus.** (A) The naturally growing plants in Beijing Badaling National Forest Park, China. (B) Flower buds (Ch01). (C) Pollinated flower (Ch02). (D) Fruits. (E) Young seeds with elaiosomes (Ch03). (F) Mature seeds with elaiosomes (Ch04). (G) Enlargement of mature seeds with elaiosomes, showing the white elaiosome is contrasted with the glossy black seed. (H) Longitudinal section of mature seed and elaiosome, showing the red-stained lipids in broken elaiosome and massive protein granulars in endosperm. (I) Diaspore (seed + elaiosome) removal of *C. majus* by ant. Bars: (A) = 10 cm, (B–D) = 1 cm, (E) = 3 mm, (F) = 2 mm, (G and I) = 1 mm, (H) = 100 μm. Abbreviation: EL, elaiosome; EN, endosperm. (Photo by Liangcheng Zhao).

plant (Fig. 1A) that is widely distributed in Europe and Asia. *C. majus* is rich in various types of isoquinoline alkaloids and is widely used against various diseases, such as ulcer, gastric cancer, oral infection, liver disorders, chronic bronchitis, asthma, and general pains, in European countries and Chinese traditional medicine (*Arora & Sharma, 2013*; *Maji & Banerji, 2015*). *C. majus* has attracted research attention not only for its medicinal value, but also for its well-known role in ant-dispersal (Fig. 1I). As a typical myrmecochorus plant, *C. majus* has been investigated in several studies on ant-seed interactions (*Gorb & Gorb, 2000*; *Fischer et al., 2008*; *Servigne & Detrain, 2008*; *Reifenrath, Becker & Poethke, 2012*). In terms of its reproductive features, *C. majus* is characterized by large yellow petals, long cylindrical capsules, and numerous small seeds (ca. 1.1–1.5 mm in length) with conspicuous coronal elaiosomes (Figs. 1C–1F). The elaiosomes (ca. 0.8–1.1 mm in length) are white and transparent, which contrasts with the glossy black color of the seeds (Figs. 1F and 1G).

The structure and development of elaiosomes in *C. majus* were briefly described by *Lisci, Bianchini & Pacini (1996)*, who showed that the elaiosomes of *C. majus* arise from the cells of the raphe and form a ventral excrescence along the seed from the hilum

and adjacent part of the micropyle to the chalazal area. Recently, one of the authors of this article (Liangcheng Zhao) conducted a detailed study on the complete ontogeny of *C. majus* elaiosomes and seeds using standard techniques for plant anatomy and histochemistry (see *Yang et al., 2015*). Different development stages have been characterized based on distinct morphological and anatomical features that can be used in genetic studies. Results indicated that the meristem of *C. majus* elaiosome is formed by periclinal divisions of the outmost layer cells of raphe in flower-bud stage (prior to fertilization); after flowering stage (post-pollination) mitotic activity ceases, and the elaiosome develops solely through very active cell growth. The mature elaiosome of *C. majus* is composed of two cell types, small basal cells and very elongated outer cells. Both cell types, especially the outer cells, contain a large amount of lipids (Fig. 1H), as verified by the red-stained oil droplets with Sudan III. In contrast, the seed primarily consists of the endosperm, which contains numerous protein granules (Fig. 1H). *Reifenrath, Becker & Poethke (2012)* performed quantitative analysis of the elaiosomes by gas chromatography and provided a detailed fatty acid composition of the *C. majus* elaiosomes. The most abundant fatty acid was determined to be linoleic acid, followed by palmitic acid, and then oleic acid; these three fatty acids comprised of nearly 96.5% of the total fatty acid content of the elaiosomes.

Although various studies have investigated the lipid composition of elaiosomes and seeds in different plant species and identified the main fatty acids, the regulatory mechanisms related to the biosynthesis of these fatty acids remain unknown. To understand the expression patterns of genes associated with fatty acid biosynthesis in elaisomes and seeds, detailed genome information is essential and indispensable. In the present study, RNA-seq technology was employed to obtain a comprehensive overview of gene expression related to fatty acid biosynthesis in *C. majus* elaiosomes and seeds for the first time. In recent years, next-generation sequencing (NGS), also known as high-throughput sequencing, has emerged as a powerful tool for profiling the whole transcriptome from non-model plants that do not have available reference genomes (*Unamba, Nag & Sharma, 2015*). In the present study, we performed whole-transcriptome profiling by NGS to capture the genomic landscape of *C. majus* elaiosomes and seeds. The analysis not only provided important insights into fatty acid biosynthesis in *C. majus* elaiosomes and seeds, but also served as a genomic resource for future evolutionary studies that focus on the convergent evolution of myrmecochorous plants.

# MATERIALS AND METHODS

## Plant materials and treatments

The sampled plants of *C. majus* were grown in Beijing Badaling National Forest Park, China, under natural conditions (Fig. 1A). To gain a comprehensive transcriptional profile of its elaiosomes and seeds, sample collection at the optimal developmental stage is crucial. Based on previous observations of *C. majus* elaiosome and seed development, four different developmental stages were defined (*Yang et al., 2015*). In the present study, we chose these four developmental stages, namely flower-bud stage (pre-fertilization) (Ch01) (Fig. 1B), flowering stage (post-pollination) (Ch02) (Fig. 1C), young seed stage (Ch03) (Fig. 1E), and mature seed stage (Ch04) (Fig. 1F) for comparative transcriptome analysis to

explore gene expression during fatty acid biosynthesis in the elaisomes and seeds of *C. majus*. Corresponding to the four stages, flower buds (with bracts, petals, and stamens removed), fertilized ovaries, young seeds (testa white to brown), and mature seeds (testa black and hard) were collected as RNA-seq materials. They were frozen in liquid nitrogen and stored at $-80\ °C$ until further use. Three samples were collected as biological repeats for each developmental stage.

## RNA isolation and sequencing

To obtain transcriptome expression profiles, RNA-seq transcriptome libraries were prepared using the TruSeqTM RNA sample preparation Kit from Illumina (San Diego, CA, USA) and five μg of total RNA. First, messenger RNA was isolated through the polyA selection method using oligo (dT) beads and fragmented in a fragmentation buffer. Next, double-stranded cDNA was synthesized using a SuperScript double-stranded cDNA synthesis kit (Invitrogen, Carlsbad CA, USA) and random hexamer primers (Illumina, San Diego, CA, USA). Then, the synthesized cDNA was subjected to end-repair, phosphorylation, and "A" base addition according to Illumina's library construction protocol. Libraries were size selected for cDNA target fragments of 200–300 bp on a 2% Low Range Ultra Agarose gel followed by polymerase chain reaction (PCR) amplification using Phusion DNA polymerase (NEB, Ipswich, MA, USA) for 15 cycles. After quantification using TBS380, the paired-end RNA-seq sequencing library was sequenced using Illumina HiSeq 4000 ($2 \times 150$ bp read length). RNA samples were named Ch01a/b/c (flower-bud stage), Ch02a/b/c (flowering stage), Ch03a/b/c (young seed stage), and Ch04a/b/c (mature seed stage). A total of 12 cDNA libraries were then constructed and sequenced using the Illumina deep sequencing platform.

## De novo assembly, annotation, differential expression analysis

The raw paired-end reads were trimmed and quality controlled by SeqPrep and Sickle with default parameters. Then, clean data were used to do de novo assembly with Trinity (*Grabherr et al., 2011*). All of the assembled transcripts were searched against the NR, Cluster of Orthologous Groups (COG), and Kyoto Encyclopedia of Genes and Genomes (KEGG) databases using BLASTX to identify the proteins that had the highest sequence similarity to the given transcripts to retrieve their function annotations; a typical cut-off *E*-value of less than $1.0 \times 10^{-5}$ was set. The BLAST2GO program was used to get Gene Ontology (GO) annotations of uniquely assembled transcripts for describing biological processes, molecular functions, and cellular components (*Conesa et al., 2005*). COG is an early database that is expected to be a useful platform for the functional annotation of newly sequenced genomes (*Tatusov et al., 2003*). The KEGG database is the primary public repository devoted specifically to pathways. Metabolic pathway analysis was performed using KEGG (*Kanehisa & Goto, 2000*), and KEGG classifications facilitate the exploration of categories of genes with specific functions. To identify differentially expressed genes (DEGs) between different samples, the expression level of each transcript was calculated according to the fragments per kilobase of the exon per million mapped reads method. RNA-Seq by expectation-maximization was used to quantify gene and isoform

**Table 1 Raw data and valid data statistics of RNA sequencing.**

| Sample_ID | Raw_reads | Clean_reads | Clean_bases | Error% | Q20%[2] | Q30%[3] | GC%[1] |
|---|---|---|---|---|---|---|---|
| Ch01a | 46,184,952 | 44,177,128 | 6,461,240,190 | 0.0138 | 97.66 | 93.3 | 45.02 |
| Ch01b | 56,008,592 | 54,268,742 | 7,954,613,129 | 0.0135 | 97.8 | 93.66 | 42.33 |
| Ch01c | 75,452,800 | 72,811,640 | 10,706,410,254 | 0.0136 | 97.78 | 93.53 | 42.81 |
| Ch01 (total) | 177,646,344 | 171,257,510 | 25,122,263,573 | 0.0136 | 97.75 | 93.5 | 43.39 |
| Ch02a | 49,629,792 | 47,787,888 | 7,035,686,679 | 0.0135 | 97.81 | 93.63 | 43.52 |
| Ch02b | 48,985,314 | 46,706,162 | 6,845,643,793 | 0.014 | 97.58 | 93.13 | 44.03 |
| Ch02c | 48,604,760 | 46,676,484 | 6,849,388,718 | 0.0137 | 97.72 | 93.43 | 44.32 |
| Ch02 (total) | 147,219,866 | 141,170,534 | 20,730,719,190 | 0.0137 | 90.7 | 93.4 | 43.95 |
| Ch03a | 46,603,464 | 45,034,614 | 6,637,326,310 | 0.0137 | 97.76 | 93.5 | 43.75 |
| Ch03b | 51,901,360 | 50,226,988 | 7,404,348,166 | 0.0135 | 97.83 | 93.67 | 43.56 |
| Ch03c | 53,968,514 | 52,416,848 | 7,726,545,508 | 0.0133 | 97.92 | 93.88 | 43.54 |
| Ch03 (total) | 152,473,338 | 147,678,450 | 21,768,219,984 | 0.0135 | 97.84 | 93.68 | 43.62 |
| Ch04a | 52,126,852 | 50,435,348 | 7,426,968,007 | 0.0135 | 97.84 | 93.67 | 43.67 |
| Ch04b | 53,366,564 | 51,854,584 | 7,647,296,324 | 0.0133 | 97.92 | 93.86 | 43.7 |
| Ch04c | 52,973,818 | 51,264,748 | 7,548,125,708 | 0.0135 | 97.84 | 93.72 | 43.48 |
| Ch04 (total) | 158,467,234 | 153,554,680 | 22,622,390,039 | 0.0134 | 97.87 | 93.75 | 43.62 |

Notes:
[1] GC%: The total number of bases G and C as a percentage of the total bases number;
[2] Q20%: The percentage of bases whose Phred value is greater than 20 accounts for the total bases;
[3] Q30%: The percentage of bases whose Phred value is greater than 30 accounts for the total bases.

abundances (*Li & Dewey, 2011*). The R statistical package software, empirical analysis of digital gene expression in R, was utilized for the differential expression analysis (*Robinson, McCarthy & Smyth, 2010*). In addition, functional-enrichment analysis, including GO and KEGG, were performed to identify which DEGs were significantly enriched in GO terms and metabolic pathways at a Bonferroni-corrected $P$-value $\leq 0.05$ compared with the whole-transcriptome background. GO functional enrichment and KEGG pathway analysis were carried out by Goatools and KOBAS (*Xie et al., 2011*).

## Quantitative real-time PCR analysis

For each sample, one μg of total RNA was used for cDNA synthesis using the Plant RNA Purification Reagent (Invitrogen, Carlsbad CA, USA). ChamQ SYBR Color qPCR Master Mix (2X) (Vazyme Biotechnology, Nanjing, China) was used for Quantitative real-time PCR (qRT-PCR). qRT-PCR was performed on Line gene 9600 plus Fluorescence quantitative PCR instrument (Hangzhou, China). Each sample contains three independent biological replicates.

## RESULTS AND DISCUSSION
### Transcriptome sequencing and de novo assembly

In this study, a total of 12 libraries were sequenced. After removing the low-quality reads and adaptor sequences, the remaining clean data comprised of approximately 61 Gb of sequence data (171 million reads from the flower-bud stage (Ch01); 141 million reads from flowering stage (Ch02); 148 million reads from young seed stage (Ch03); and 154 million

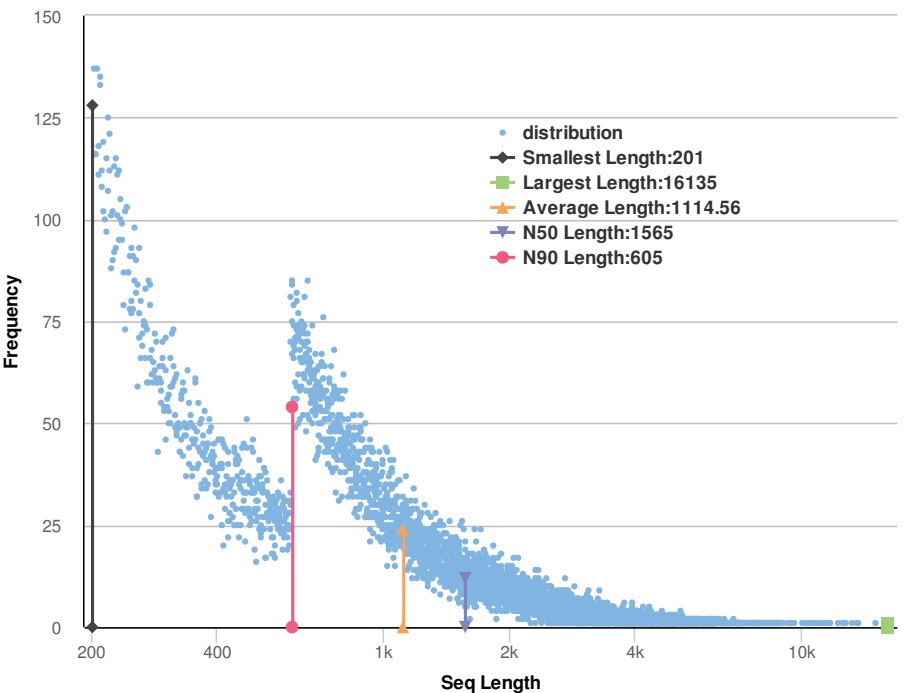

**Seq Length Distribution**

**Figure 2 Statistic of trinity assembly results.** N50: Assembled unigenes by length from large to small, and the length of it is half the length of the unigenes to the total (bp). N90: Assembled unigenes by length from large to small, and the length of it is 90% the length of the unigenes to the total (bp).

reads from mature seed stage (Ch04)). The results are shown in Table 1. De novo assembly of the clean reads was performed using Trinity to generate a reliable integrated reference sequence database (https://doi.org/10.6084/m9.figshare.8044838.v1, File 1). Some reads comprise of a contig, and some contigs comprise of a unigene. We obtained a total of 63,064 unigenes (https://doi.org/10.6084/m9.figshare.8044838.v1, File 2) for all studied samples; these unigenes had an average length and N50 value of 1,114.56 bp and 1,565 bp, respectively. After sorting the assembled unigenes from large to small, the N50 value was calculated as half of the total length of the unigenes. The size distribution of the assembled transcripts is shown in Fig. 2. As shown in Table 1 and Fig. 2, the sequencing data were of sufficiently high quality for subsequent analyses. Therefore, gene expression data from *C. majus* elaiosomes and seeds were reliable.

## Functional annotation and classification of unigenes

To verify the potential functions of the 63,064 assembled unigenes, we performed sequence similarity BLASTx searches against the NR (NCBI protein nonredundant), COG, GO, and KEGG protein databases with $E$-values $< 1e^{-5}$. After alignment, gene annotation, functional classification, and metabolic pathway assignment, a total of 30,143 (47.80%) of the unigenes were successfully annotated (https://doi.org/10.6084/m9.figshare.8044838.v1, File 3), which indicated that their functions were relatively conserved and diverse.
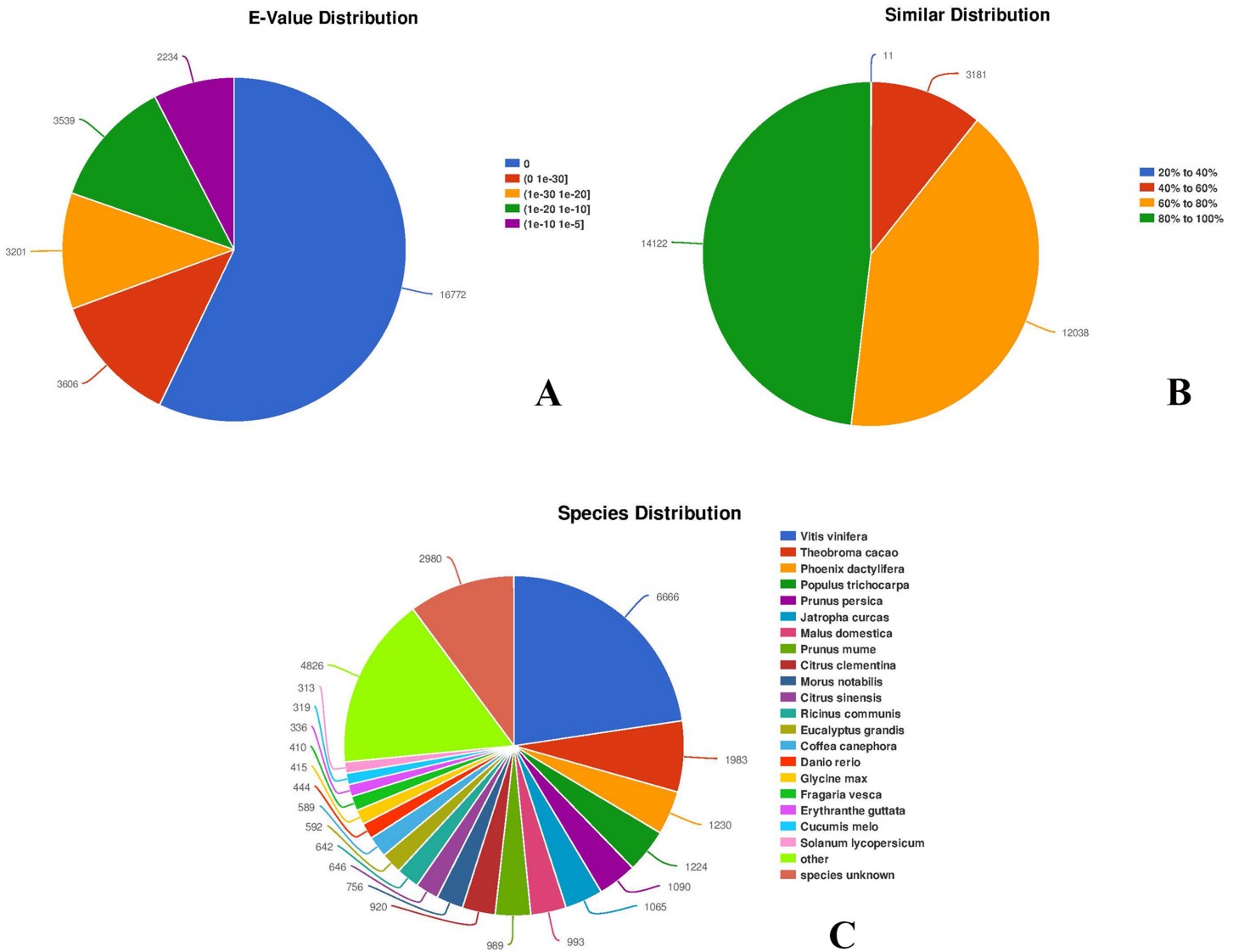

**Figure 3 Characteristics of sequence homology of unigenes against NR database.** (A) $E$-value distribution of BLAST hits for each unigene with an $E$-value cut-off of $1.0 \times 10^{-5}$. (B) Similarity distribution of the top BLAST hits for each unigene. (C) Species distribution of the top BLAST hits.

## Characterization of unigene sequencing

Among all the 30,143 annotated unigenes, 29,352 (97.38%) were mapped to the NR database. In addition, the distribution of $E$-values of the hits in the NR database indicated that 20,378 unigenes, representing more than half of the mapped sequences (69.43%), showed significant homology ($E$-value $< 1.0 \times 10^{-30}$), while 8,973 unigenes (about 30.57%) had $E$-values ranging from $1.0 \times 10^{-30}$ to $1.0 \times 10^{-5}$ (Fig. 3; https://doi.org/10.6084/m9.figshare.8044838.v1, File 3). Smaller $E$-values distribution (Fig. 3A) indicated a higher reliability of hits. The similarity distribution showed that the majority of the mapped 26,160 unigenes (89.13%) had similarity values >60%, whereas only 3,192 (10.87%) had similarity values <40% (Fig. 3B). To identify species specificity, individual unigenes were

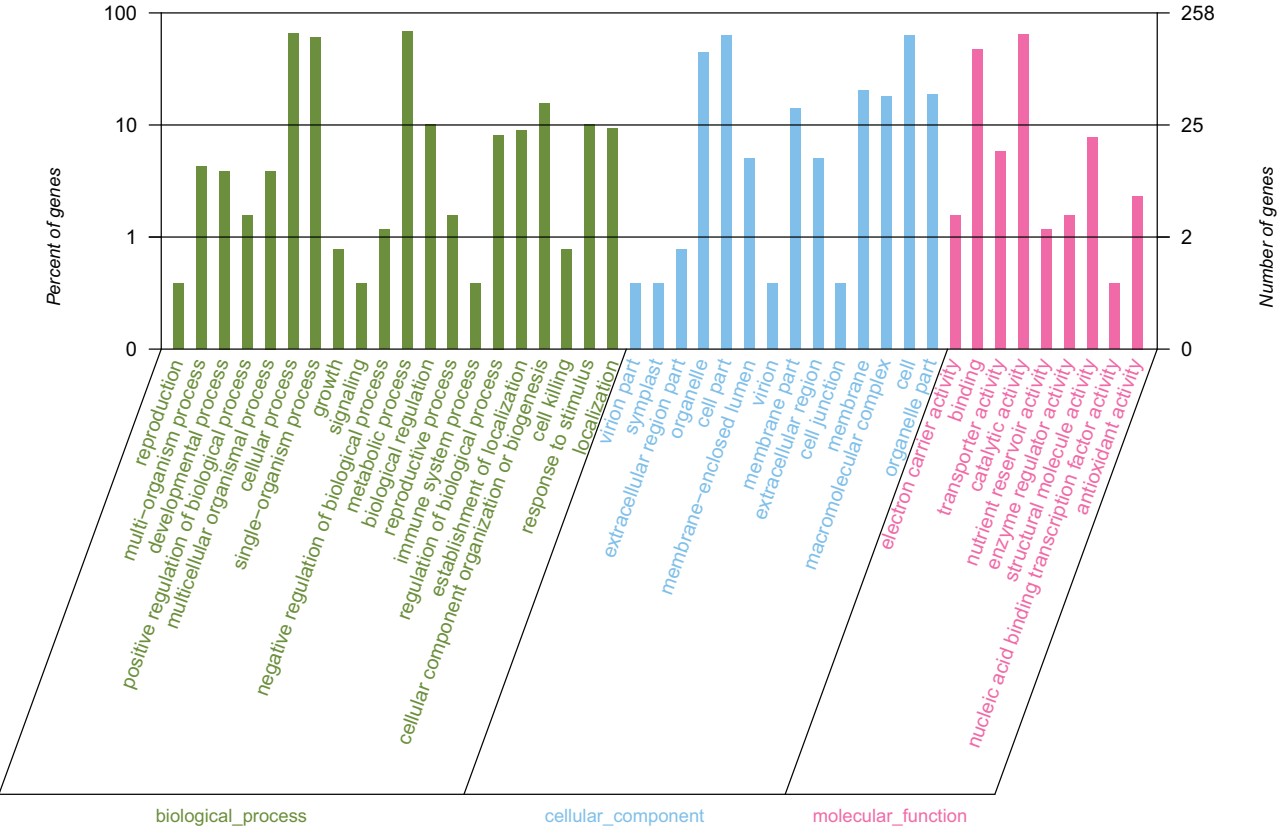

**Figure 4  GO classification of assembled unigenes.** The abscissa represents the secondary classification terms of GO, the left ordinate represents the percentage of unigenes contained in the secondary classification, the right ordinate represents the number of unigenes. These three colors represent three categories: green represents the biological process, blue represents the cellular component, and red represents the molecular function.

annotated based on the highest BLAST scores against the NR database. Among flowering plants, 6,666 (22.71%) unigenes showed significant homology to *Vitis vinifera*, followed by *Theobroma cacao* (1,983 unigenes, 6.76%), *Phoenix dactylifera* (1,230 unigenes, 4.19%), *Populus trichocarpa* (1,224 unigenes, 4.17%), *Prunus persica* (1,090 unigenes, 3.71%), and *Jatropha curcas* (1,065 unigenes, 3.63%) (Fig. 3C). A total of 2,980 unigenes, representing around 10.15% of all unigenes, did not match the NR database, and thus provided novel genomic information on *C. majus*. Some of these unigenes are likely to be untranslated regions or non-coding RNAs.

## Functional classification by gene ontology

By conducting GO functional classification, a total of 258 unigenes were classified into one or more terms and divided into 43 functional groups, belonging to three major categories (Fig. 4). In the biological process category, the majority of the assigned proteins were involved in metabolic processes (177 unigenes, 68.60%), followed by genes involved in cellular processes (169 unigenes, 65.50%) and single-organism processes (156 unigenes, 60.47%). Under the molecular function category, the mapped unigenes were categorized into nine groups, out of which genes were predominantly associated with catalytic activity (165 unigenes, 63.95%) and binding (123 unigenes, 47.67%), which exceeded the total

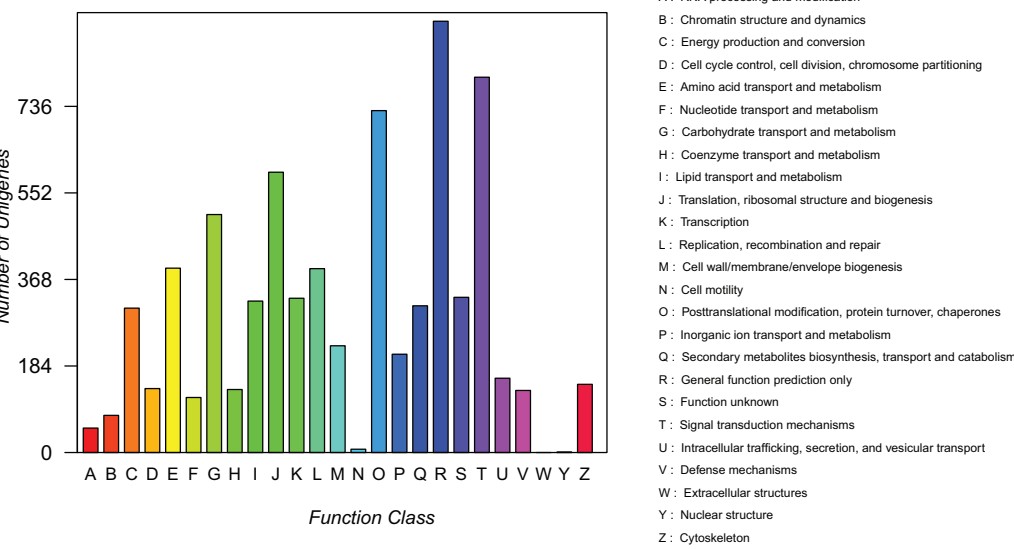

**COG Function Classification**

A : RNA processing and modification
B : Chromatin structure and dynamics
C : Energy production and conversion
D : Cell cycle control, cell division, chromosome partitioning
E : Amino acid transport and metabolism
F : Nucleotide transport and metabolism
G : Carbohydrate transport and metabolism
H : Coenzyme transport and metabolism
I : Lipid transport and metabolism
J : Translation, ribosomal structure and biogenesis
K : Transcription
L : Replication, recombination and repair
M : Cell wall/membrane/envelope biogenesis
N : Cell motility
O : Posttranslational modification, protein turnover, chaperones
P : Inorganic ion transport and metabolism
Q : Secondary metabolites biosynthesis, transport and catabolism
R : General function prediction only
S : Function unknown
T : Signal transduction mechanisms
U : Intracellular trafficking, secretion, and vesicular transport
V : Defense mechanisms
W : Extracellular structures
Y : Nuclear structure
Z : Cytoskeleton

**Figure 5 COG function classification of assembled unigenes.** Each column represents the functional classification of COG (Capital letters A~Z are used to indicate the specific meaning, as indicated on the right), and the height of the column is the ordinate of the number of unigenes with this function.

number of the other terms. The cellular component category was divided into 14 groups, in which the largest assignments were found to be in the cell and cell part. There were 164 unigenes in the cell and cell part (63.57%), followed by organelles (115 unigenes, 44.57%), membrane (52 unigenes, 20.16%), and membrane parts (36 unigenes, 13.95%). The above results suggested that the development of elaisomes and seeds in *C. majus* are closely associated with enzyme-catalyzed reactions, cell structure formation, and metabolic processes.

## Functional classification by clusters of orthologous groups

To predict the potential function of genes and evaluate the completeness of the transcriptome libraries, all assembled unigenes were searched against the COG database. A total of 7,323 mapped unique sequences were clustered into 25 functional categories (Fig. 5). Among them, the largest category was "general functions prediction only" (917, 12.52%), followed by "signal transduction mechanisms" (798, 10.90%), "posttranslational modification, protein turnover, and chaperones" (727, 9.93%), and "translation, ribosomal structure, and biogenesis" (596, 8.14%). Thus, we inferred that these genes play an important role in ribosome formation, protein synthesis, and signal transduction during the development of elaisomes and seeds in *C. majus*. Notably, 322 unigenes (4.40%) were classified under the "lipid transport and metabolism" category, which could provide useful information for further studies on lipid metabolism in *C. majus* elaisomes and seeds.

## Biological pathways identification by KEGG

To identify the biological pathways that were actively involved in lipids accumulation in *C. majus* elaisomes and seeds, all annotated unigenes were mapped to the reference

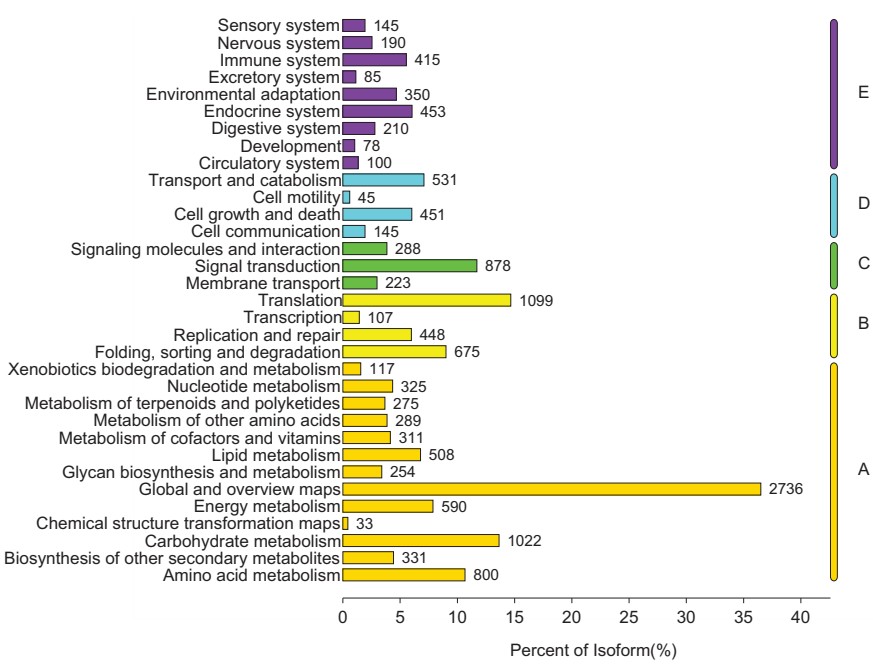

**Figure 6 Distribution of unigenes in different KEGG pathways.** Genes were divided into five clusters. A, Metabolism; B, genetic information processing; C, environmental information processing; D, cellular process; E, organismal systems.               

pathways in the KEGG. The analysis identified a total of 11,540 unigenes that were assigned to five clusters and distributed across 373 KEGG pathways (Fig. 6). Most of the unigenes were grouped under the metabolism cluster (A). Within this cluster, the majority of unigenes were assigned under the "global and overview maps pathway" (2,736, 23.71%), followed by "carbohydrate metabolism" (1,022, 8.85%), "amino acid metabolism" (800, 6.93%), and "energy metabolism" (590, 5.11%) categories. The relative abundance of unigenes involved in "amino acid metabolism" pathways is probably related to the biosynthesis of massive storage protein in the *C. majus* endosperm. Notably, in the "energy metabolism" cluster, some pathways were found to be closely related to the lipid metabolism. A total of 508 unigenes, including glycerophospholipid metabolism (205, 40.35%), glycerolipid metabolism (108, 21.26%), fatty acid degradation (100, 19.69%), biosynthesis of unsaturated fatty acids (93, 18.31%), fatty acid elongation (75, 14.76%), fatty acid biosynthesis (74, 14.57%), ether lipid metabolism (68, 13.39%), sphingolipid metabolism (54, 10.63%), linoleic acid metabolism (44, 8.66%), steroid biosynthesis (42, 8.27%), arachidonic acid metabolism (21, 4.13%), steroid hormone biosynthesis (16, 3.15%), synthesis and degradation of ketone bodies (12, 2.36%), and primary bile acid biosynthesis (5, 0.98%) pathways were involved in these pathways (Fig. 7).

## Analysis of DEGs at four developmental stages

Analysis of the fatty acid compositions of *C. majus* seeds and elaiosomes showed that the elaiosome-bearing seeds predominantly contained the monounsaturated fatty acid, linoleic acid (C18:2) (77.9%), followed by the monounsaturated fatty acid, oleic acid

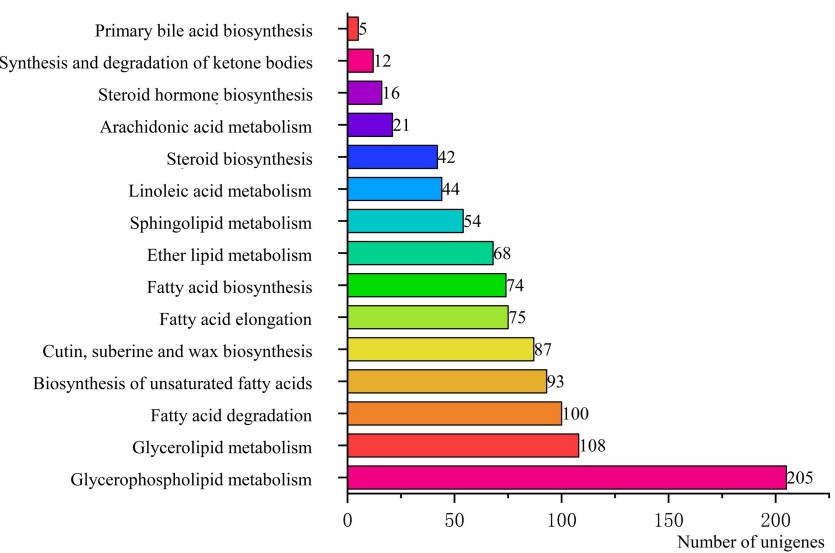

**Figure 7 Pathways in "lipid metabolism" group and number of their unigenes.** Each column represents the lipid metabolism pathways of KEGG, and the height of the column is the ordinate of the number of unigenes with this pathway.

(C18:1) (16%) (*Reifenrath, Becker & Poethke, 2012*). Furthermore, saturated fatty acids accounted for only 6% of the composition, and most of the saturated fatty acids were identified as palmitic acid (C18:1) (3.5%). However, in only elaiosomes, the most abundant fatty acid also was linoleic acid (37.2%), followed by palmitic acid (34.8%), and oleic acid (24.5%). Previous developmental and histochemical observations (*Yang et al., 2015*) indicated that the elaiosome meristem of *C. majus* was formed during the flower-bud stage (Ch01), while elaiosome cell growth and enlargement primarily occurred from the flowering stage (Ch02) to the young seed stage (Ch03); the maturation of the seed and elaiosome occurred from the young seed stage to the mature seed stage (Ch04). Therefore, to obtain insight into the regulation of the biosynthesis of saturated fatty acids and unsaturated fatty acids in elaiosomes and seeds, we analyzed the DEGs of the lipid metabolism pathways in Ch02 compared to Ch01, Ch03 compared to Ch02, and Ch04 compared to Ch03.

## Identification of unigenes related to fatty acid biosynthesis

From the fatty acids biosynthesis pathway (Path: ko00061) and the biosynthesis pathway of unsaturated fatty acids (Path: ko01040) at the four developmental stages of *C. majus* elaisomes and seeds (File S1), we identified 41 significantly DEGs. These DEGs are shown in File S2 and the corresponding genetic heat map is shown in Fig. 8. Four unigenes encoding the enzymes malonyl CoA-ACP transacylase (*fabG*) (*Toomey & Wakil, 1966*), acyl-CoA oxidase (*ACOX1*) (*Oaxaca-Castillo et al., 2007*), acyl carrier protein desaturase (*DESA1*) (*Kachroo et al., 2007*), and β-ketoacyl–acyl carrier protein synthase II (*fabF*) (*Magnuson, Carey & Cronan, 1995*) were found to be upregulated in Ch02 compared to Ch01. By contrast, 24 unigenes encoding the enzymes acetyl-coA carboxylase (*Accase*) (*Bilder et al., 2006*), acyl coenzyme A synthetase (*ACSL*) (*Black et al., 1992*), *DESA1*,

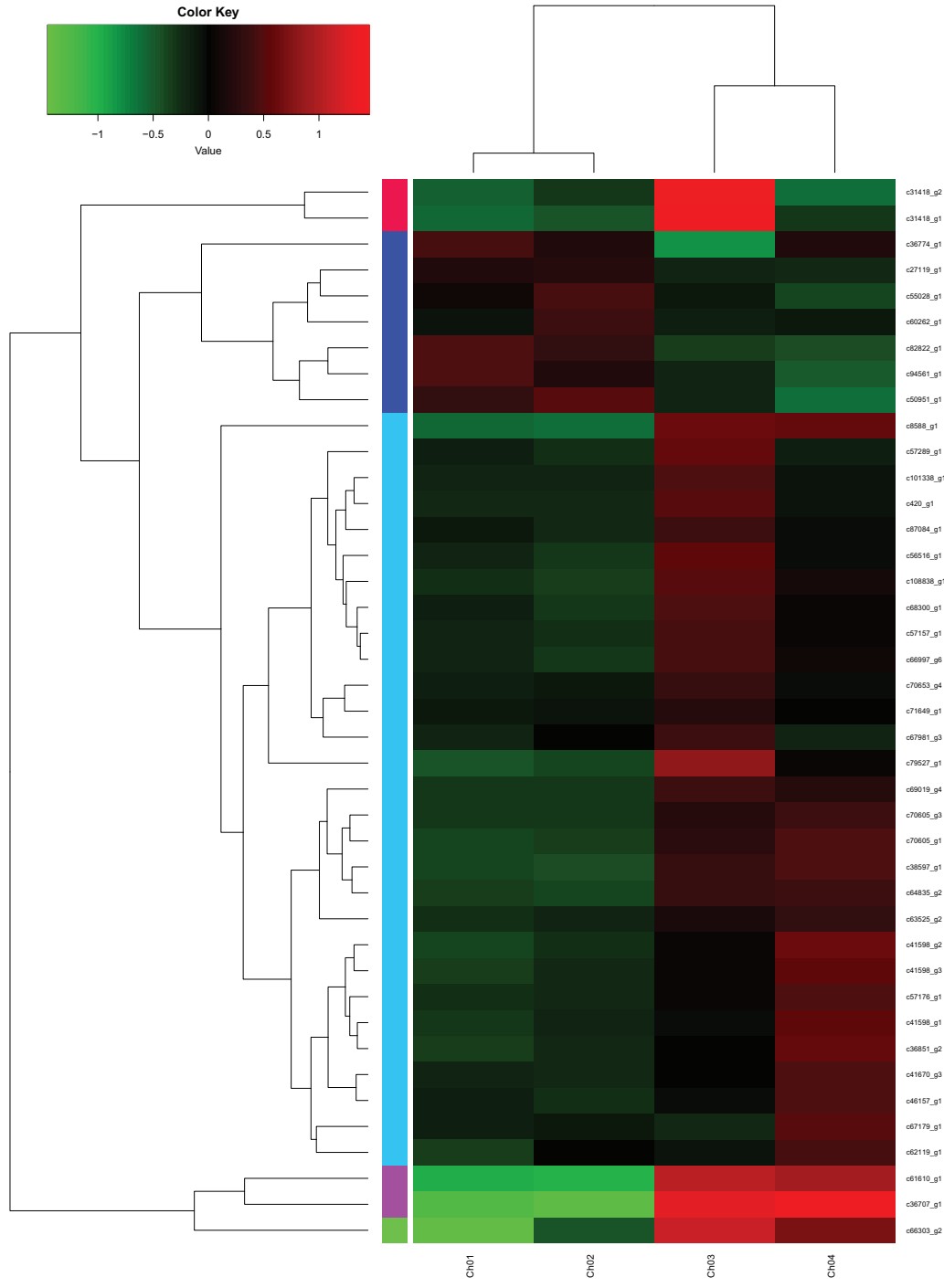

**Figure 8 DEGs from the fatty acid biosynthesis pathway and biosynthesis of unsaturated fatty acid pathway diagram.** The locus tag and enzyme code shown in fatty acid biosynthesis pathway and biosynthesis of unsaturated fatty acid pathway are described in a heat map.

acyl-carrier-protein S-malonyltransferase (*fabD*) (*Magnuson et al., 1992*), *fabF*, *fabG*, enoyl-[acyl-carrier-protein] reductase (*fabI*) (*Bergler et al., 1996*), β-hydroxyacyl-acyl carrier protein dehydratases (*fabZ*) (*Heath & Rock, 1996*), acyl-lipid omega-6 desaturase
**Table 2 DEGs in fatty acid biosynthesis metabolic pathway for different developmental stages.**

| KEGG pathway | Gene | Gene ID | Gene name | Enzyme | Ch01 vs Ch02 DEGs number | | Ch02 vs Ch03 DEGs number | | Ch03 vs Ch04 DEGs number | |
|---|---|---|---|---|---|---|---|---|---|---|
| | | | | | UR[1] | DR[2] | UR | DR | UR | DR |
| path:ko00061 | Accase (accA) | K01962 | acetyl-CoA carboxylase | EC 6.4.1.2 | 0 | 2 (0) | 1 (1) | 1 (0) | 1 (0) | 1 (1) |
| path:ko00061 | Accase (accB) | K02160 | acetyl-CoA carboxylase | EC 6.4.1.2 | 2 (0) | 1 (0) | 3 (3) | 0 | 2 (0) | 1 (1) |
| path:ko00061 | Accase (accC) | K01961 | acetyl-CoA carboxylase | EC 6.4.1.2 | 0 | 1(0) | 1(1) | 0 | 0 | 1(1) |
| path:ko00061 | Accase (accD) | K01963 | acetyl-CoA carboxylase | EC 6.4.1.2 | 1 (0) | 0 | 1 (1) | 0 | 0 | 1 (1) |
| path:ko00061 | Accase (ACACA) | K11262 | acetyl-CoA carboxylase | EC 6.4.1.2 | 1 (0) | 0 | 0 | 1 (0) | 1 (0) | 0 |
| path:ko00061 | fabD | K00645 | [acyl-carrier-protein] S-malonyltransferase | EC 2.3.1.39 | 0 | 1 (0) | 1 (1) | 0 | 0 | 1 (1) |
| path:ko00061 | fabF | K09458 | 3-oxoacyl-[acyl-carrier-protein] synthase II | EC 2.3.1.179 | 1 (1) | 4 (0) | 5 (4) | 0 | 1 (0) | 4 (3) |
| path:ko00061 | fabG | K00059 | beta-ketoacyl reductase | EC 1.1.1.100 | 1 (1) | 2 (0) | 2 (1) | 1 (1) | 2 (1) | 1 (1) |
| path:ko00061 | DESA1 | K03921 | acyl carrier protein desaturase | EC 1.14.19.2 | 1 (1) | 1 (0) | 2 (2) | 0 | 0 | 2 (0) |
| path:ko00061 | fabZ | K02372 | β-hydroxyacyl-acyl carrier protein dehydratases | EC 4.2.1.59 | 1 (0) | 0 | 1 (1) | 0 | 0 | 1 (0) |
| path:ko00061 | fabI | K00208 | enoyl-[acyl-carrier-protein] reductase | EC 1.3.1.10 | 0 | 1 (0) | 1 (1) | 0 | 0 | 1 (1) |
| path:ko00061 | FATA | K10782 | acyl-ACP thioesterase A | EC 3.1.2.14 | 0 | 1 (0) | 1 (1) | 0 | 0 | 1 (0) |
| path:ko00061 | FATB | K10781 | acyl-ACP thioesterase B | EC 3.1.2.14 | 1 (0) | 2 (0) | 2 (1) | 0 | 2 (1) | 1 (0) |
| path:ko00061 | ACSL | K01897 | acyl coenzyme A synthetase | EC 6.2.1.3 | 4 (0) | 4 (0) | 7 (1) | 1 (0) | 5 (2) | 3 (2) |

Notes:
[1] UR: Upgraded unigenes number, data in bracket represents significantly DEGs.
[2] DR: Downgraded unigenes number, data in bracket represents significantly DEGs.

(Delta-12 desaturase) (*FAD2*) (*Kainou et al., 2006*), acyl-ACP thioesterases (*FatA/B*) (*Salas & Ohlrogge, 2002*), and stearoyl-CoA desaturase (Delta-9 desaturase) (*SCD*) (*Stukey, McDonough & Martin, 1990*) were clearly upregulated in Ch03 compared to Ch02. However, in Ch04 compared to Ch03 identified that out of the 24 previously mentioned unigenes, 13 genes encoding *Accase, ACSL, fabD, fabF, fabG, fabI,* and *SCD* were downregulated, while the 11 remaining genes were not differentially expressed. The above results indicated that there are more relevant DEGs in Ch02 vs Ch03 compared to Ch01 vs Ch02 and Ch03 vs Ch04. Based on the observations of *C. majus* elaiosome development (*Yang et al., 2015*), the remarkable cell growth of the elaiosome also occurred during the flowering stage to the young seed stage (Ch02 to Ch03). Therefore, we inferred that the biosynthesis of saturated fatty acids and unsaturated fatty acids in *C. majus* elaisomes might primarily occur during this period.

## Identification of unigenes related to the biosynthesis of saturated fatty acid

In the fatty acid biosynthesis metabolic pathway, a comparison of the DEGs in Ch02 compared to Ch01 (Table 2; File S3A) showed that most genes that are primarily involved
in fatty acid biosynthesis were not significantly expressed during the flowering stage (Ch02), and only a few genes were upregulated. By contrast, the gene expression analysis in Ch03 compared to Ch02 (Table 2; File S3B) revealed that the majority of genes were significantly expressed during the young seed stage (Ch03). Most of these expressed genes, which encode Accase, *ACSL*, *DESA1*, *fabD*, *fabF*, *fabG*, *fabI*, *fabZ*, and *FatA/B*, were significantly upregulated in Ch03. On the other hand, a gene expression analysis in Ch04 compared to Ch03 (Table 2; File S3C) revealed that almost all genes were significantly downregulated during the mature seed stage (Ch04). The above results indicated that the biosynthesis of saturated fatty acid in *C. majus* elaisomes and seeds also mainly occurred during the flowering stage to the young seed stage. However, some unigenes that encode *FabG*, exhibited inverse expression patterns. The above-mentioned genes were found to be downregulated in Ch03 compared to Ch02 but were upregulated in Ch02 compared to Ch01 and in Ch04 compared to Ch03 (Table 2). These genes are potentially involved in fatty acid synthesis through negative feedback signaling.

The enzymes *FatA/B* and *DESA1*, which are the key enzymes involved in palmitic acid synthesis (see File S3), are particularly noteworthy, because palmitic acid is the most abundant saturated fatty acid in *C. majus* elaisomes but is present in very low quantities in the corresponding seeds. *FatA/B* was slightly upregulated in Ch02 compared to Ch01 and significantly upregulated in Ch03 compared to Ch02, but was significantly downregulated in Ch04 compared to Ch03. *DESA1* was significantly upregulated in both Ch02 compared to Ch01 and Ch03 compared to Ch02, but significantly downregulated in Ch04 compared to Ch03. The above findings indicate that the biosynthesis of palmitic acid in *C. majus* elaisomes occurs from the flower bud stage to the flowering stage (Ch01 to Ch02) but primarily occurs from the flowering stage to the young seed stage. *Hughes, Westoby & Jurado (1994)* compared the fatty acid compositions of 12 plant species of elaisomes and seven orders of insects and showed that palmitic acid levels in elaisomes were particularly similar to those in insects. However, because *C. majus* was not included in Hughes's study, whether the high palmitic acid content in *C. majus* elaisomes is similar to that in the ants that disperse the *C. majus* seeds remains unknown and needs further study.

## Identification of unigenes related to the biosynthesis of unsaturated fatty acid

Figures 8 and 9 show that significantly upregulated genes were identified in Ch03 compared to Ch02 and were involved in three processes in the biosynthesis of unsaturated fatty acid metabolic pathway. The first process is the conversion of palmitic acid (C16:0) to stearic acid (C18:0), which involves genes encoding the enzyme, *fabG*. The second process is the conversion of stearic acid (C18:0) to oleic acid (C18:1, $\Delta$9), which involves genes encoding the enzymes, *SCD* (Delta-9 desaturase) and *DESA1*. The third process is the conversion of oleic acid (C18:1, $\Delta$9) to linoleic acid (C18:2, $\Delta$9$\Delta$12), which involves genes encoding the enzyme, *FAD2* (Delta-12 desaturase). The above findings showed that some palmitic acid was converted into stearic acid (saturated fatty acid), and the latter was subsequently desaturated to form oleic acid and linoleic acid.

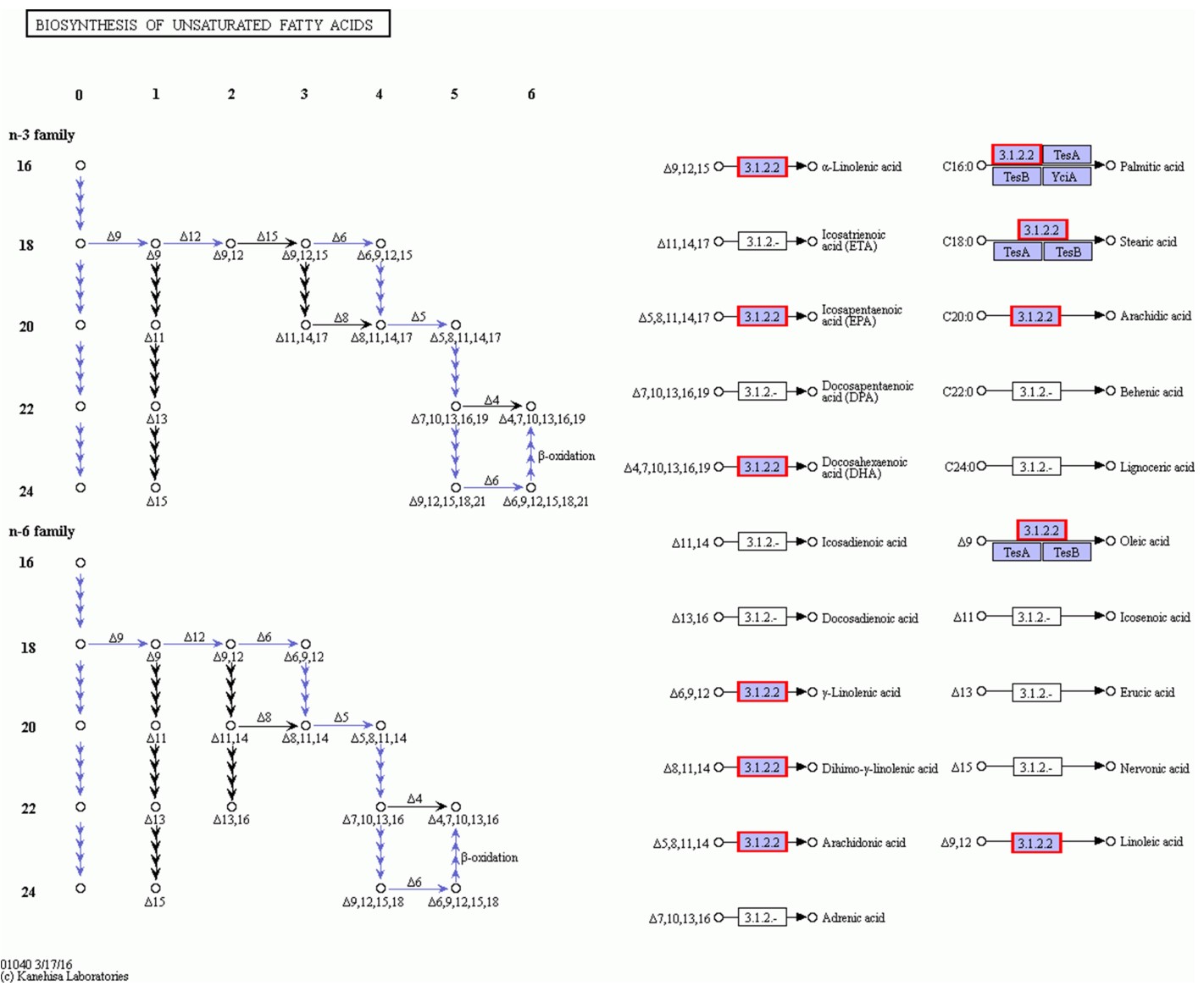

**Figure 9 The detail RNA degradation database in biosynthesis of unsaturated fatty acid pathway.** The red border are the genes that can annotate on these gene products.

Linoleic acid is a poly-unsaturated fatty acid that cannot be synthesized by insects de novo and is thus required to be incorporated in their diet (*Dadd, 1973*). Previous studies have suggested increasing linoleic acid levels by stimulating the diaspore-carrying behavior of the ants (*Gammans, Bullock & Schönrogge, 2005*). The high linoleic acid content in both elaiosomes and seeds of *C. majus* indicated that the whole diaspores (seed + elaiosome) were probably used to attract ants that contain high levels of (essential) nutrients. In addition, oleic acid was shown to promote ant removal behavior (*Marshall, Beattie & Bollenbacher, 1979*; *Skidmore & Heithaus, 1988*; *Brew, O'Dowd & Rae, 1989*; *Hughes, Westoby & Jurado, 1994*). The higher oleic acid content in *C. majus* elaiosomes and the lower oleic acid content in their corresponding seeds suggested that oleic acid

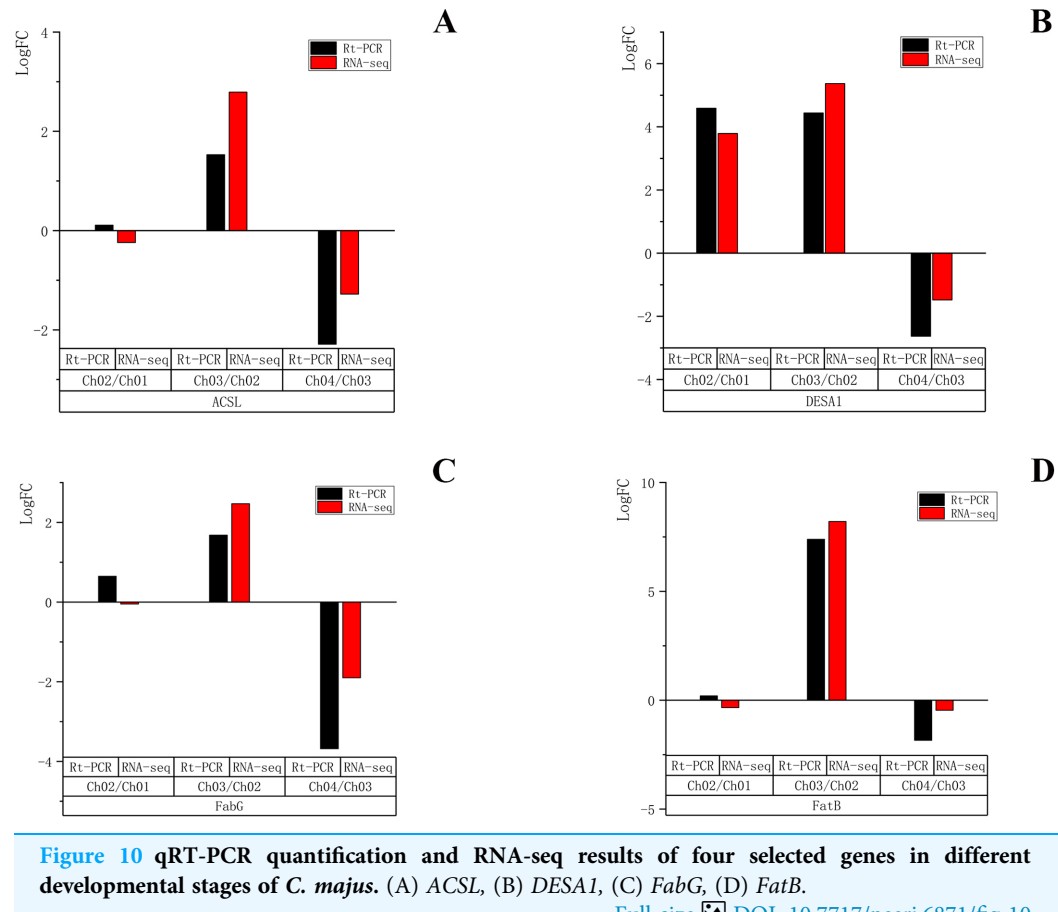

**Figure 10 qRT-PCR quantification and RNA-seq results of four selected genes in different developmental stages of *C. majus*.** (A) *ACSL*, (B) *DESA1*, (C) *FabG*, (D) *FatB*.

in *C. majus* might act as the primary stimulating substance to induce the collection of its diaspores by ants, as previously reported in other plant species.

## Validation of unigenes by quantitative real-time PCR

To validate the reliability of the RNA-seq results, four unigenes encoding *ACSL*, *DESA1*, *fabG*, and *FatB* were randomly selected for qRT-PCR validation at the four different developmental stages. These four genes were all related to the fatty acid biosynthesis metabolic pathway and biosynthesis of unsaturated fatty acid metabolic pathway. The qRT-PCR results (File S4) showed that the expression patterns of the four selected genes were very similar to those of RNA sequencing data in Ch03 compared to Ch02 and in Ch04 compared to Ch03 (Fig. 10). In Ch01 compared to Ch02, the expression pattern of *DESA1* by qRT-PCR was consistent with that from RNA sequencing (Fig. 10B), and *ACSL*, *fabG*, and *FatB* showed slight differences (Figs. 10A, 10C and 10D). It was acceptable that some differences in direct comparison between qRT-PCR and RNA-seq results would occur due to biases in the library preparation for RNA-seq, different normalization approaches, and other technical biases (*Li et al., 2010*; *Wei, Chung & Zhao, 2011*). qRT-PCR results of the selected genes were generally consistent with the RNA sequencing results, indicating that the RNA-seq results were reliable.

## CONCLUSIONS

*Chelidonium majus* is a well-known ant-dispersal plant with numerous elaiosome-bearing seeds. In this study, we performed whole-transcriptome profiling of *C. majus* elaiosomes and seeds by Illumina NGS sequencing, and used times series RNA-seq to study the expression patterns of genes associated with fatty acid biosynthesis in them for the first time. A total of 63,064 unigenes were generated and 41 significantly DEGs involved in lipid metabolism pathways were identified. By analysis and comparison of the DEGs at the four developmental stages of the elaiosomes and seeds, we confirmed that the fatty acid biosynthesis in the elaiosome is consistent with its cellular growth. The candidate genes involved in the biosynthesis of three primary fatty acids in elaiosomes and seeds were also identified. Our research not only provided important insights into fatty acid biosynthesis in *C. majus* elaiosomes and seeds, but also served as a genomic resource for future evolutionary studies that focus on the convergent evolution of myrmecochorus plants.

## ACKNOWLEDGEMENTS

The authors would like to acknowledge Beijing Badaling National Forest Park for support with plant materials. We also thank our colleagues for discussions and advice on this study. We are also grateful to two anonymous reviewers for improving our manuscript.

### Funding

This work was supported by the the National Natural Science Foundation of China (No. 31370213). The funders had no role in study design, data collection and analysis, decision to publish, or preparation of the manuscript.

### Grant Disclosure

The following grant information was disclosed by the authors:
National Natural Science Foundation of China: 31370213.

### Competing Interests

The authors declare that they have no competing interests.

### Author Contributions

- Jiayue Wu conceived and designed the experiments, performed the experiments, analyzed the data, contributed reagents/materials/analysis tools, prepared figures and/or tables, authored or reviewed drafts of the paper, approved the final draft.
- Linlin Peng performed the experiments, analyzed the data, authored or reviewed drafts of the paper.
- Shubin Dong conceived and designed the experiments, analyzed the data, authored or reviewed drafts of the paper.
- Xiaofei Xia conceived and designed the experiments, authored or reviewed drafts of the paper.

- Liangcheng Zhao conceived and designed the experiments, analyzed the data, contributed reagents/materials/analysis tools, prepared figures and/or tables, authored or reviewed drafts of the paper, approved the final draft.

## DNA Deposition

The following information was supplied regarding the deposition of DNA sequences:

The DNA sequences described here are accessible via NCBI GenBank submission number: SUB4838035, BioProject accession number: PRJNA507335.

Raw reads are available in the Sequence Read Archive database (SRR8255587–SRR8255598).

## Data Availability

Wu, Jiayue; Zhao, Liangcheng (2019): Trinity results of Chelidonium majus. figshare. Dataset. https://doi.org/10.6084/m9.figshare.8044838.v1.

## Supplemental Information

Supplemental information for this article can be found online at http://dx.doi.org/10.7717/peerj.6871#supplemental-information.

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
