# Peer review of "Transcriptome analysis of Chelidonium majus elaiosomes and seeds provide insights into fatty acid biosynthesis"

_PeerJ, doi:10.7717/peerj.6871_

## Round 0.1 · original submission · Major Revisions

Dear Jiayue and Liangcheng,

Thank you for your submission to PeerJ. Please address the issues raised by the reviewers.

Reviewer 1 ·

Basic reporting

The manuscript "Transcriptome analysis of Chelidonium majus elaiosomes and seeds provide insights into fatty acid biosynthesis" presents reliable and interesting data on the fatty acid biosynthesis in Chelidonium majus elaiosomes and seeds using comparative transcriptomic approach. The English is professional and well used throughout the text, however the authors did not avoid some repetitions, as e.g. in line 344 the word "further" used twice "...further studies are required to further verify....".
The authors deal with the interesting issue of myrmecochory and provide sufficient background to understand the results.

Experimental design

The manuscript is within the scope of PeerJ. Research question is well-defined and fills an important gap of knowledge on the fatty acid biosynthesis in C. majus elaiosomes and seeds.
However, there is inconsistency in the description of developmental stages of C. majus between sections "Plant Material and Treatments" and "RNA isolation and Sequencing" - in line 140 it is stated that Ch04 stage is called "mature seed stage" and in line 160 the same stage is named "early mature seed stage". Please make it consistent and explain what is the background behind your division on stages (maybe the small table could help to organise it in a more comprehensible way).
Additional comments:
- the legend of Figure 1 should contain symbols of the distinguished developmental phases (Ch01, Ch02, Ch03, Ch04), which should be added to this figure legend in the relevant places,
- line 176, "fragments per kilobase of the exon per million mapped reads" should be abbreviated as "FPKM", not as "FRKM".

Validity of the findings

The data presented are reliable and the authors prove this fact in several places of the manuscript, however in some points of the manuscript inconsistencies could be found:
- in line 198 it is stated that "The results are shown in Table 1"; however when looking at Table 1 mentioned data in preceding lines (196-198) are not present therein, it contains other values, so it must be reviewed and potentially expaned to contain relevant data,
- lines 201, 208 - is the number of 63,064 unigenes found in the study the total number for all studied samples?
- lines 220-221 - the authors state that their data is reliable, and it is true, but this fact is just the simple consequence of presented data and does not have to be mentioned several times? Please consider to transfer these data (presented on Fig. 3A) to supplementary material.
- the data presented on Figures 8-10 are very complex and their description between the lines 294-356 seems to be too descriptive. Please consider preparing additional table(s) to better present the main differences between dev. stages and selected genes and enzymes, what could better support final conlusions. Figures 8 and 9 could be then considered to be included as supplementary information.

Additional points:
- line 193, "de novo" should be written in italics throughot the manuscript; the word "Assembly" should be written here beginning with small letter "assembly",
- supplementary information (lines 425-427) should contain titles of each tables, and supplementary tables themselves should contain the title and the legend on one of the panels inside the file to better present the results and make them more understandable.

Reviewer 2 ·

Basic reporting

1. Unigene is a Trinity specific term. The authors should define and describe it first. The general audiences wouldn't know what is a unigene.
2. Please check the grammar and typos throughout the text, especially the figure and table captions!

Experimental design

The experimental design is sound, carefully designed time series expression data with biological replicates, and qRT-PCR. However, not enough data is shared to ensure the result can be replicated. For example, the script for finding DEG genes; the script/tool for functional-enrichment analysis. For those are not code, e.g., the unigenes from Trinity, the authors can also provide the program output files as supplementary files. The bottom line is to share as much as possible to ensure replicative research.

Validity of the findings

The text says saturated fatty acids and unsaturated fatty acids but figure 8 caption mentioned only unsaturated fatty acids

"The above results indicated that the biosynthesis of saturated fatty acids and unsaturated fatty acids in C. majus elaisomes and seeds primarily occurred during the flowering stage to the young seed stage (Ch02 to Ch03), consistent with the remarkable cell growth of the elaiosome observed during this period (Yang et al., 2015)."

I think this is an overstatement. The above results indicated there are more relevant DEG genes in Ch02 vs Ch03 compared to Ch01 vs Ch02 and Ch03 vs Ch04. However, whether that indicates the biosynthesis occurred primarily during Ch02 to Ch03 requires more explanations or evidence.

"Therefore, we inferred that the high palmitic acid content in C. majus elaiosomes is similar to that in the ants that disperse the C. majus seeds. However, further studies are required to further verify the findings in C. majus and the ants."

Does the Hughes et al. (1994) include C. majus and the ants? If yes, just cite their study. If not this statement is speculation without any support from the text.

Additional comments

This study provides valuable research and data to study the expression patterns of genes associated with fatty acid biosynthesis in elaisomes and seeds of C. majus. And I tend to believe it is the first study to use times series RNA-Seq to study this problem since no relative results are compared against. And I think the author should make it clear in the text. The results are significant and consistent with prior knowledge. However, I do believe to make this work more impactful, the authors should also provide
1. The Raw RNA-Seq data (can upload to SRA), and qRT-PCR result.
2. The intermediate files, such as Trinity results and analyses scripts.

---

## Round 0.2 · accepted · Accept

Dear Jiayue and Liangcheng

I am pleased to inform you that your manuscript is accepted by PeerJ. Your article will be edited by the editor and you will receive a list of tasks. Please pay attention to it.

Thank you.

# "I did note some pointers to COG and KEGG pathways, and mention of GO annotation; however, I did not see a table with these listings, or a way to make connections. It is not very helpful that the assignments were generated when the corresponding sequence alignments of source assignments can be compared. I would suggest that the raw data be made available in a open data repository. The manuscript is in an acceptable format; however, added value would be recognized if the effort were to deposit the data somewhere. Journal manuscripts are often scanned by text-mining software that locates and extracts core data elements, like gene function. Adding standard ontology terms, such as the Gene Ontology (GO, geneontology.org) or others from the OBO fountry (obofoundry.org) can enhance the recognition of your contribution and description. This will also make human curation of literature easier and more accurate."

If you are able, then you can incorporate these comments while in Production. #

Reviewer 1 ·

Basic reporting

no comment

Experimental design

no comment

Validity of the findings

no comment

Additional comments

Hereby I would like to congratulate the authors and repeat the opinion from my previous review, that the manuscript "Transcriptome analysis of Chelidonium majus elaiosomes and seeds provide insights into fatty acid biosynthesis" presents reliable and interesting data and fills an important gap of knowledge on the fatty acid biosynthesis in C. majus elaiosomes and seeds. The authors have answered on all my questions and corrected the manuscript accordingly. Therefore I recommend to publish the manuscript in PeerJ as is.

Reviewer 2 ·

Basic reporting

The new manuscript has addressed the previous comments.

Experimental design

The new manuscript has addressed the previous comments.

Validity of the findings

The new manuscript has addressed the previous comments.

Additional comments

The new manuscript has addressed the previous comments. This study is in scope with PeerJ publication interest. The result and its RNA-seq data fill in some gaps of knowledge about fatty acid biosynthesis in C. majus elaiosomes and seeds. I recommend accepting this manuscript.